# Co-Design for Enhancing Flood Resilience in Davao City, Philippines

**Mamoru Miyamoto** [1,*], **Daiki Kakinuma** [1], **Tomoki Ushiyama** [1], **Abdul Wahid Mohamed Rasmy** [1], **Masaki Yasukawa** [2], **Della Grace Bacaltos** [3], **Anthony C. Sales** [4], **Toshio Koike** [1] and **Masaru Kitsuregawa** [5]

[1] International Centre for Water Hazard and Risk Management, Public Works Research Institute, Tsukuba 3058516, Japan; kakinuma-d977bt@pwri.go.jp (D.K.); ushiyama55@pwri.go.jp (T.U.); abdul@pwri.go.jp (A.W.M.R.); t-koike@pwri.go.jp (T.K.)
[2] Earth Observation Data Integration and Fusion Research Initiative, The University of Tokyo, Tokyo 1538505, Japan; yasukawa@iis.u-tokyo.ac.jp
[3] Davao del Sur State College, Digos City 8008, Philippines; bacaltosdella@gmail.com
[4] Department of Science and Technology Region XI, Davao City 8000, Philippines; dr.acsales@region11.dost.gov.ph
[5] National Institute of Informatics, Tokyo 1018430, Japan; kitsure@tkl.iis.u-tokyo.ac.jp
* Correspondence: mmiyamoto@pwri.go.jp; Tel.: +81-29-879-6779

**Abstract:** Enhancing flood resilience, including the development of social capacity and early warning systems, in addition to structural measures, is one of the key solutions to mitigating flood damage, which will be more intensified in the future due to climate change. This study was conducted to develop a comprehensive methodology for enhancing flood resilience by improving society-wide disaster literacy under the governance formed through the active participation of all levels of stakeholders in Davao City, Philippines. Specifically, the development of the Online Synthesis System for Sustainability and Resilience, which integrates different disciplines, and the fostering of Facilitators, whose role is to interlink the science community and society, were implemented in a co-designing manner by the collective governance body. The development of basin- and barangay-scale hydrological models realized real-time flood forecasting and climate change impact assessment to identify intensified flood risk under the future climate. Co-designed e-learning workshops were held to foster about thirty Facilitators and help them produce twenty-one risk communication plans and workshop designs for fourteen barangays considering geographic, demographic, economic, and social features that they can utilize for public dissemination related to climate change adaptation to the target audiences in society. This paper presents a practical method to enhance flood resilience, demonstrating that the synthesis of science-based knowledge and human resource development can fill the gaps between the science community and society.

**Keywords:** OSS-SR; facilitator; flood resilience; disaster literacy; community-based; e-learning

## 1. Introduction

Given the floods that have occurred around the world in recent years, flood risk management is undoubtedly an urgent and major social issue. The World Meteorological Organization has reported that the number of weather-related disasters that have hit the world has increased five-fold over the past 50 years [1]. The trend is due to the increased water vapor in the atmosphere, exacerbating extreme rainfall and deadly flooding, as a result of climate change. More than 90% of the mortalities due to weather-related disasters have occurred in developing countries, though the death number has decreased significantly over the last 50 years. This fact implies that strengthening resilience, such as the development of social capacity and early warning systems, is one of the key solutions to averting water-related disasters, which will be more intensified in the future due to climate change.

In international settings on disasters, the term "resilience" is quoted with the UNISDR's definition in the Hyogo Framework for Action 2005–2015 [2]: "The capacity of a system, community or society potentially exposed to hazards to adapt, by resisting or changing in order to reach and maintain an acceptable level of functioning and structure." However, Fisher (2015) pointed out that more than 70 definitions of resilience existed across the scientific literature [3] and that they roughly fall into two categories: "the ability of a system to bounce back after stress" and "the capacity of social-ecological systems to adapt or transform in response to unfamiliar, unexpected and extreme shocks". Kerri (2020) systematically reviewed the literature on flood risk management and clarified the position and importance of flood risk management in the interdisciplinary understanding of resilience [4]. Efforts have also been made to verify and compare the effect of resilience by devising different approaches, for example, developing frameworks, indices, and models to quantitatively assess flood resilience [5–11]. For the quantitative assessment of flood resilience, while hydrological models or indicators are employed, a unified approach has not yet been established because adjustments to the study area are often necessary. In addition to the importance of resilience assessment, an emphasis should also be placed on how to enhance flood resilience from the practical perspective of flood risk mitigation. Some case studies of qualitative approaches have attempted to enhance the flood resilience of specific communities through particular methods of risk communication or awareness-raising activities using web-based systems [12–15]. However, the existence of various stakeholders and communities in society and the segmentalization of scientific disciplines nowadays make it difficult to realize the actual enhancement of the flood resilience of society as a whole. Morrison et al. (2017) presented five themes—stakeholder engagement; policy effectiveness; research in practice; tools; and frameworks—by investigating literature on governance for flood resilience [16]. They also concluded that research on governance for flood resilience lacks integration of themes, and methods of mitigating this lack of integration are poorly studied. Hence, the development of holistic methods for substantially enhancing flood resilience at all levels of stakeholders and communities is an indispensable research theme for society and has not been addressed in previous studies.

This study aims to develop a comprehensive methodology to enhance flood resilience in society as a whole by improving society-wide disaster literacy based on the governance in which all stakeholders engage. This study, therefore, demonstrates the importance of co-design by the science community and all levels of society in Davao City, Philippines, for beneficial and sustainable achievements. Figure 1 shows the overall structure that this paper uses to formulate a method to enhance society-wide flood resilience based on co-design among relevant stakeholders. We begin with the localization of a general concept of flood resilience enhancement in the study area. In line with the localized concept design, we developed an integrated system, Online Synthesis System for Sustainability and Resilience (OSS-SR), for consilience and fostered Facilitators, which are detailed in the Methodology presented in Section 3. We also produced an institutional design of interactive risk communication by Facilitators to fill gaps between the science community and society for the enhancement of flood resilience.

## 2. Study Area

Davao City has 182 barangays, the smallest political unit in the Philippines [17]. Its population growth rate from 1990 to 2010 was 2.4%, with the total population rising to 1,449,296 in 2010. The population density was 5.9 persons per hectare in 2010. Among the urban barangays, the population density reached 43 persons per hectare, whereas the population density among the rural barangays was estimated to be 1.5 persons per hectare. Agricultural land accounts for the largest share in the land use classification, with 30% of the total land area of Davao City. The second highest land type, forest areas, occupies 16.4%, and urban use 5.4%.

The Davao River is one of the 18 major rivers in the Philippines and is prioritized for master planning because of its socio-economic importance. Severe floods often occur

in January, June, and July. In 2011, more than 10,000 families were affected by floods in Davao City. Out of its 130 km² built-up area, 10.9 km² is considered highly flood-prone, and 42.1 km² is classified as residential areas. Figure 2 shows the study area of 3644 km², covering eight river basins surrounding Davao City, including the Davao River basin. As part of the Hydrology for the Environment, Life and Policy (HELP)-UNESCO Program, the HELP Davao Network, which has been one of the most active local initiatives, was established for better informing complex decisions and hard choices concerning the wise management and use of water.

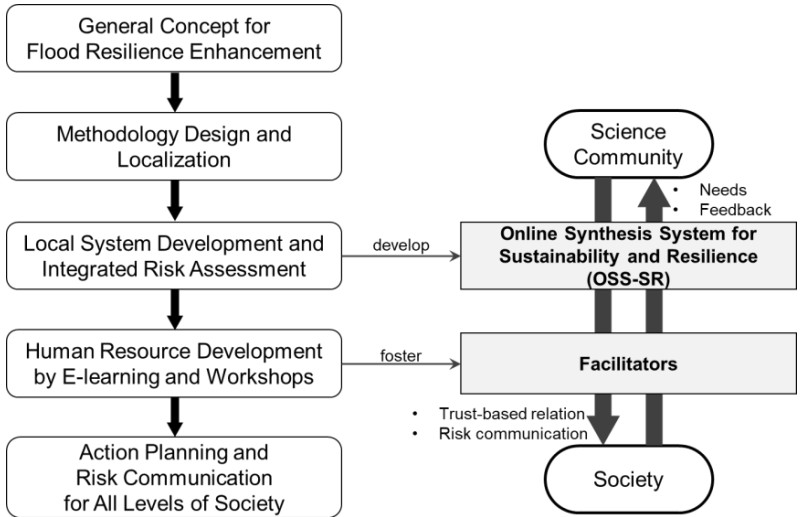

**Figure 1.** Structure of this study for formulating a method to enhance society-wide flood resilience based on co-design among relevant stakeholders.

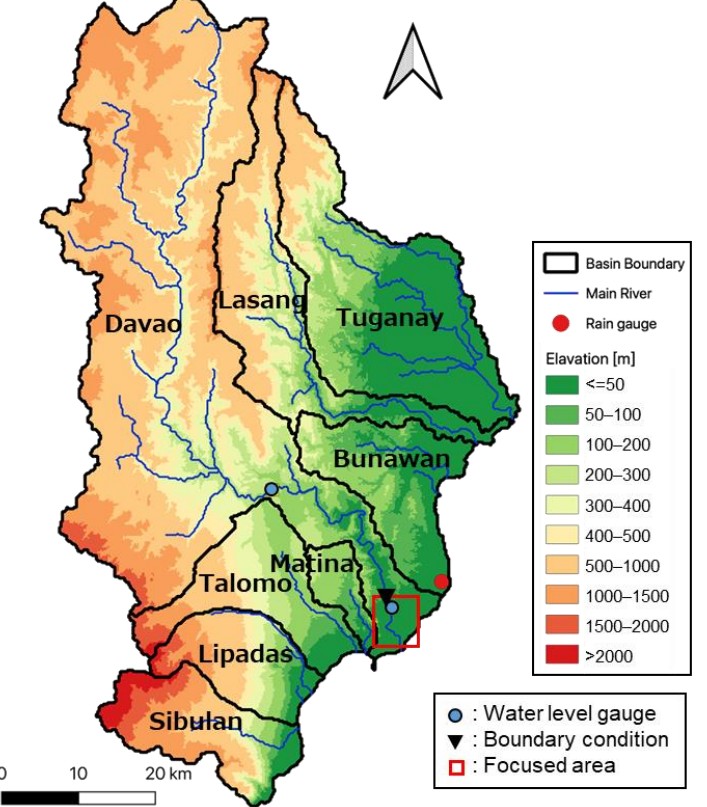

**Figure 2.** Eight river basins around Davao City, including the Davao River basin. The locations of rainfall gauges, water level gauges, boundary conditions, and the focus area are also shown.

## 3. Methodology

### 3.1. General Concept for Disaster Resilience Enhancement

Major challenges that science faces in order to improve disaster literacy and enhance disaster resilience at all levels of society include how to integrate different disciplines of science and how to fill the gaps with local communities. In its recommendation published in 2020 [18], the Committee on International Cooperation for Promoting Science-Based Disaster Risk Reduction of the Science Council of Japan addressed necessary elements to enhance disaster resilience: consilience and human resources. Methodologies to strengthen consilience and human resources are described concerning developing the OSS-SR, which is designed to integrate knowledge, information, experience, awareness, and models under different disciplines, and to foster catalytic Facilitators, whose role is to interlink the scientific community and the local society. This study employs the methodologies in the recommendation and implemented them as an end-to-end science covering cutting-edge science and community practices to enhance resilience to water-related disasters in the Philippines.

During the eighth meeting of High-level Experts and Leaders Panel on Water and Disasters (HELP) on 1 November 2016 [19], the International Flood Initiative (IFI) discussed flood resilience under climate change by organizing a side event, and the HELP-IFI Jakarta Statement based on discussions in the side event was adopted. The statement highlights the interdisciplinary and transdisciplinary partnership to establish a platform as part of a national platform for facilitating dialogue among all stakeholders from national, local, and community levels [20]. With this statement and the recommendation in the outcome document of High-Level Panel on Water released in 2018 [21], Platforms on Water Resilience and Disasters (PLATFORM), which are institutional frameworks involving all levels of stakeholders to enhance the resilience of the whole society for water-related disasters, have been established since 2017 in the Philippines, Sri Lanka, Myanmar, Indonesia, and other countries. PLATFORM also focuses on scientific themes of data integration, flood forecasting, contingency planning, climate change impact, economic assessment, and agricultural productivity, as well as capacity development, in order to contribute to the creation of social benefits in such areas as decision-making, policy-making, local practice, and investment principles. The governance of PLATFORM helps social implementations to develop OSS-SR and foster Facilitators, as well as support them in starting cooperation with Davao City in the Philippines.

### 3.2. Development of OSS-SR for Davao City

The general concept for enhancing disaster resilience has been localized in Davao City by accumulating discussions with stakeholders and considering climatic, geopolitical, and social aspects. Figure 3 shows the concept design and OSS-SR for Davao City, highlighting two focuses of real-time flood forecasting and climate change impact assessment. OSS-SR also has an e-learning function to provide users with ten introductory lectures, examinations, and four classes of hands-on training, widely covering issues in climate change, flood management, and disaster risk reduction. The e-learning function enables fostering Facilitators, who translate scientific knowledge and information for actors in society, including decision-makers, policymakers, government officers, DRR practitioners, civil society organizations, the private sector, the media, and local communities.

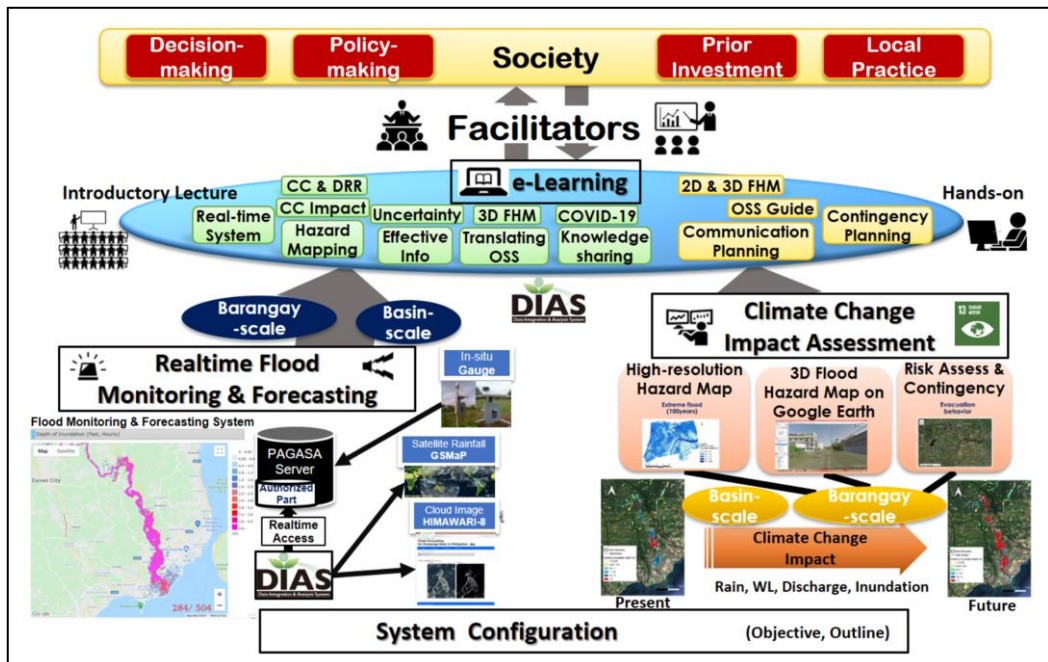

**Figure 3.** Concept design of the Online Synthesis System for Sustainability and Resilience (OSS-SR) and the Facilitator fostering for Davao City in the Philippines.

### *3.3. Hydrological Model Development*

We developed two scales of hydrological inundation models: a basin-scale model with 1 km resolution for creating an understanding of the overall picture of the basin and a barangay-scale model with 40 m resolution to investigate the detailed conditions of critical infrastructure in the focus area. The focus area and locations of gauging stations and boundary conditions for the barangay-scale model are also shown in Figure 2.

The basin-scale model employed the WEB-RRI model, a water and energy budget-based Rainfall-Runoff-Inundation model applicable to wet- and dry-climate river basins [22]. The model was calibrated by the river discharge of the annual simulation of 2002 at the Lacson station, the upstream water-level station. The model was also validated by the river discharge of the annual period and the biggest flood event of 2008, as shown in Figures 4 and 5, respectively. Due to the low availability of long-term hydrometeorological gauge data, we employed satellite-based rainfall data, Climate Hazards Group InfraRed Precipitation with Station data (CHIRPS) [23], for calibration and validation. Although there are still challenges in calibration and validation because of the uncertainty of the rainfall data themselves, Figure 4 shows a reasonable agreement between the observed and simulated discharge. The event-based comparison provided in Figure 5 also shows rough agreement, though the Nash–Sutcliffe efficiency [24] is a modest value of 0.32. While the cause of this relatively low index value has not been clearly identified, it may be due to the insufficient reliability of the satellite-based rainfall input. The barangay-scale model employs the RRI model, which focuses on floods and simultaneously simulates rainfall-runoff and flood inundation processes on a 2-D basis at a river basin scale [25].

These models are connected by inputting the river discharge simulated by the basin-scale model into the barangay-scale model as a boundary condition. The synthesis of these two models with different spatial scales in the Data Integration Analysis System (DIAS) realized real-time local flood forecasting and a climate-change impact assessment. DIAS is a system for the creation of an information storage infrastructure for applications of public benefits and the deepening of scientific knowledge in the areas of the climate and water cycle for application in fisheries, agriculture, and biodiversity management, particularly through the linking of information across disciplines [26].

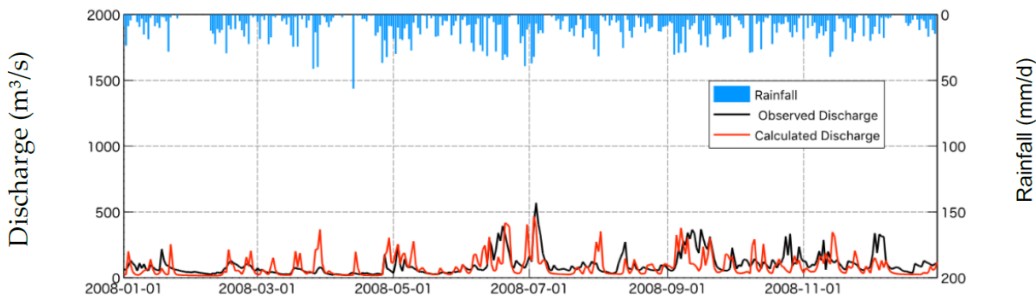

**Figure 4.** Validation results of river discharge at the Lacson station with CHIRPS, satellite-based rainfall.

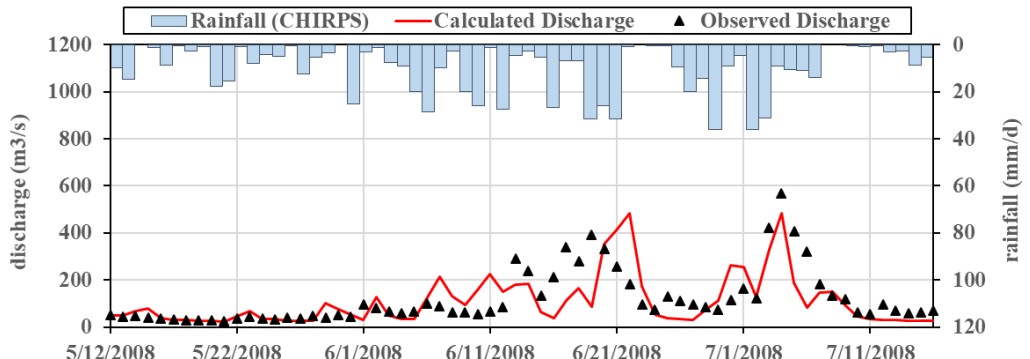

**Figure 5.** Event-based validation of river discharge at the Lacson station with CHIRPS, satellite-based rainfall during the annual maximum flood.

### 3.4. Co-Design of E-Learning Workshops

E-learning workshops were proposed as an effective means of fostering Facilitators and undertaken after co-designing the structure of the workshops among relevant stakeholders of academia and all levels of society. PLATFORM fulfilled its role as the governance body and was able to gather cooperation from various stakeholders. Through the co-designing of the e-learning workshops, one of the most important yet controversial points was how to gather candidates for Facilitators from different disciplines and sectors in society. Based on the past efforts and future direction related to disaster resilience, we came up with four selection criteria, as shown in Table 1: direct disciplines, a good mix of sciences, representation from different levels of governance, and local initiative. In line with the results of the co-designing effort, two e-learning workshops were held in April 2021 and January 2022.

**Table 1.** Criteria for gathering Facilitator candidates from different disciplines and sectors in the society.

|  | Category | Expertise |
| --- | --- | --- |
| Criteria 1 | Direct disciplines | disaster risk reduction, flood management, meteorology, climate change, integrated water resources management |
| Criteria 2 | Good mix of sciences | natural science, engineering, social science, ICT |
| Criteria 3 | Representation from different levels of governance | city/municipality office, national government, community leader, private sector, media, academia |
| Criteria 4 | Local initiative | civil society organizations, non-governmental organizations |

## 4. Results

### 4.1. Development of OSS-SR

OSS-SR for Davao City, which was developed in DIAS, stores and displays observed data, such as satellite-based rainfall, GSMaP [27], and cloud images of Himawari-8. Advances in the integration of remote sensing technologies strengthen the effectiveness of flood monitoring functions, contributing to timely and proper decision-making [28]. Observed data not only help to understand spatial rainfall distribution but can also be used as input data for simulation models. OSS-SR also illustrates the extent of inundation and depth from the results of real-time forecasting and climate-change impact assessment conducted using the basin- and barangay-scale models with rainfall data stored in DIAS. Furthermore, OSS-SR delivers ten materials of introductory lectures and four materials of hands-on training. In addition to the information from numerical models and analysis, on-site information and experiences can also be archived and shared among users with photos and texts on local information maps. Figure 6 shows an example of a local information map. The accumulation of local information, fed by residents from various local spots in the area, contributes to the advancement of science.

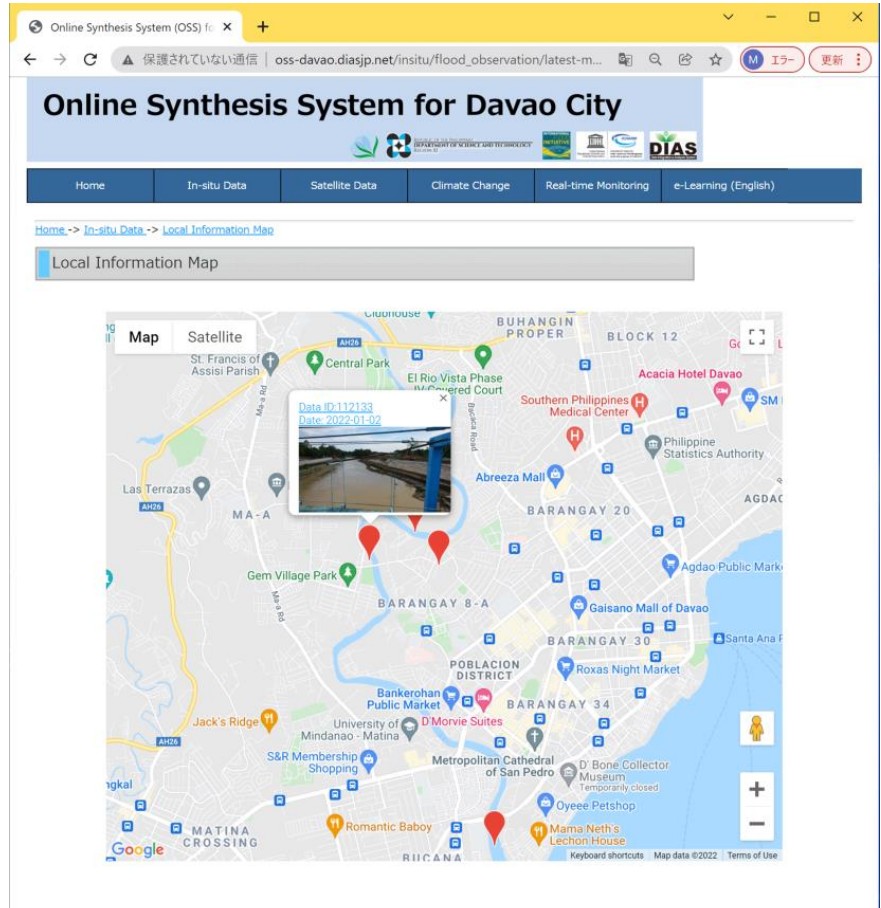

**Figure 6.** Example window of local information map in the OSS-SR for Davao City.

### 4.2. Climate Change Impact Assessment

Tropical cyclones are likely to become more intense in the future, as evidenced by recorded tropical cyclones in several cities and municipalities in Mindanao such as Davao City, which used to be known as a typhoon-free metropolis. Cabrera and Lee (2018) reported the necessity of immediate actions of decision-makers to develop a community-based disaster risk plan under future climate in Davao Oriental [29].

To assess the climate change impact in the Davao region, we employed a GCM output of MRI-AGCM 3.2H developed by the Meteorological Research Institute of Japan, because this paper focuses on communities' understanding of hazard estimation due to past and future floods for saving lives, not the facility planning based on a statistical approach with ensemble GCM outputs. MRI-AGCM 3.2H was selected because of the advantage of its high resolution and four kinds of multi-SST. The RCP8.5 Scenario of MRI-AGCM 3.2H was dynamically downscaled to 5 km by the WRF model with no convection scheme. Since the probability result of past climate after downscaling showed gaps with observed data, CHIRPS was used to perform bias correction based on monthly factors of cumulative probabilities for daily rainfall, excluding the top 0.5%, as was applied to the Pampanga River basin in the Philippines [30]. Regarding the downscaled GCM outputs, the worst extreme events in the past climate from 1979 to 2003 and the future climate from 2075 to 2099 were identified in terms of 24 h maximum rainfall and used for the impact assessment of the hazard estimation.

The rainfall data sets of the worst extreme events in the past and future climate were firstly input to the basin-scale WEB-RRI model. Figure 7 shows the river discharge of the worst flood events for the past and future climate, simulated by the WEB-RRI basin-scale model. Although the response of river discharge depends on the spatiotemporal distribution of extreme rainfall, the peak discharge of the worst event in the future is about 2.5 times that of past events. These time series of river discharge were applied to the barangay-scale model as the boundary condition at the upstream end. Figure 8 compares the basin-scale inundation maps of the worst extreme events in the past and future climate. Due to the difference in rainfall intensity and peak discharge, the inundation extent and depth are expected to be more significant in the future. The area more than 0.3 m deep under the future climate was more than four times as large as that under the past climate. Figure 9 compares the barangay-scale inundation maps of the worst extreme events in the past and future climate. The barangay-scale inundation maps also illustrate the significance of the future climate in terms of flood extent and depth with the spatial distribution of flood risk indicated at a resolution that enables users to learn to what risk level each critical infrastructure, evacuation facility, and evacuation route is exposed. Table 2 summarizes the comparisons between the past and future climate in terms of 24 h maximum rainfall, peak discharge, inundation extent, and inundation volume. The inundation area and volume in the table were calculated for the areas more than 0.3 m deep. All items indicate an increase in future flood disaster risk.

**Table 2.** Comparisons between the past and future climate from the aspects of 24 h maximum rainfall, peak discharge, inundation extent, and inundation volume.

| | | Past | Future | Future/Past |
|---|---|---|---|---|
| | 24 h maximum rainfall (mm) | 114.0 | 198.1 | 1.7 |
| | Peak discharge ($m^3$/s) | 1029.8 | 2564.7 | 2.5 |
| Whole basin | Inundation area ($km^2$) | 35.0 | 144.1 | 4.1 |
| | Inundation volume ($10^6$ $m^3$) | 176.7 | 456.2 | 2.6 |
| Focus area | Inundation area ($km^2$) | 2.8 | 7.1 | 2.5 |
| | Inundation volume ($10^6$ $m^3$) | 2.7 | 10.7 | 4.0 |

OSS-SR-based e-learning also provided information about a climate change scenario of MRI-AGCM output, uncertainties, and bias correction. Importantly, the results presented in Table 2 only concern the worst event cases in one GCM output. Flood inundation does not always occur as simulated because the magnitude and spatiotemporal distribution of floods vary from event to event. Hence, OSS-SR also has the function of real-time flood forecasting to contribute to reducing the risk of damage for a specific upcoming flood.

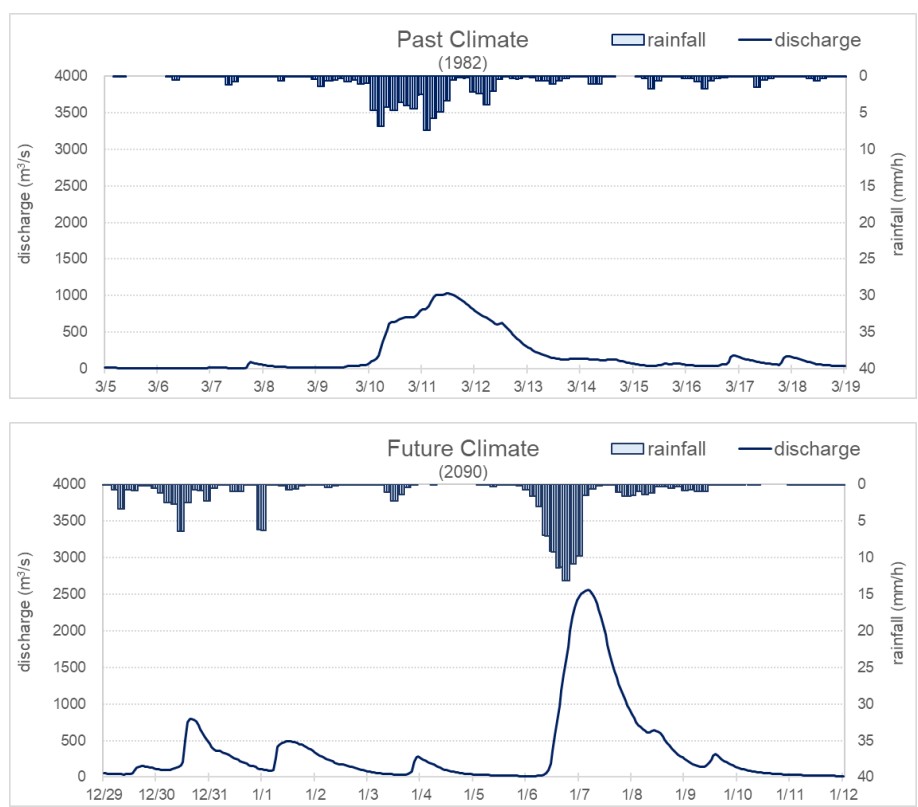

**Figure 7.** Results of river discharge at the upstream end of the focus area during the worst flood events for the past (**upper**) and future (**lower**) climate, simulated by the basin-scale model. The peak discharge in the future is about 2.5 times that of the past. These time series of river discharge were applied to the barangay-scale model as the boundary condition.

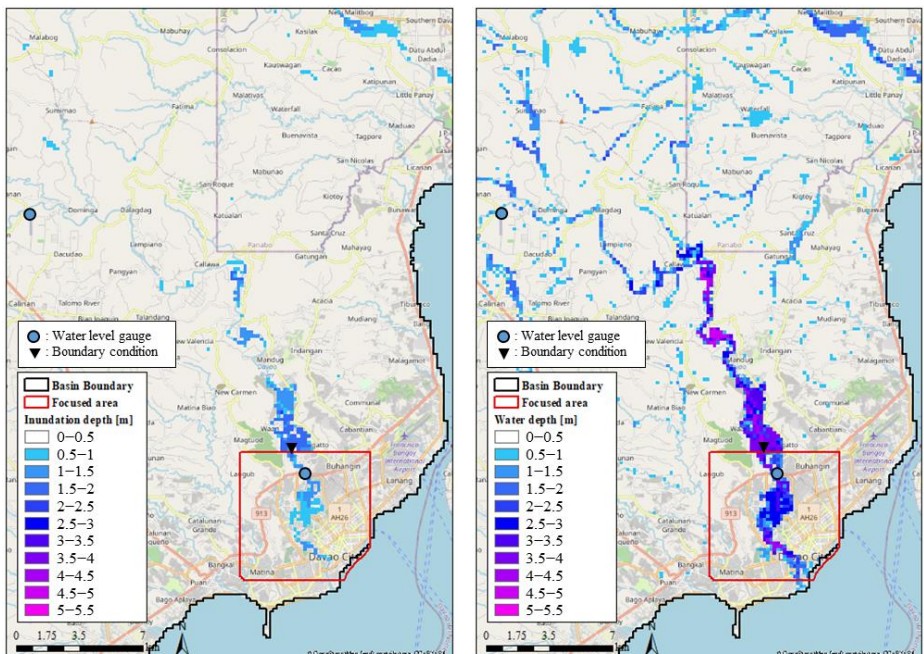

**Figure 8.** Basin-scale inundation maps of the worst extreme events in the past (**left**) and future (**right**) climate. The inundation extent in the future is more than 4 times that of the past.

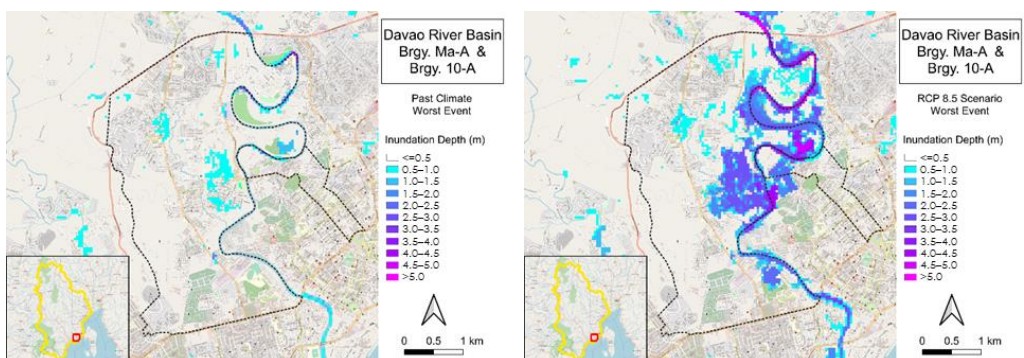

**Figure 9.** Barangay-scale inundation maps of the worst extreme events in past (**left**) and future (**right**) climate. The inundation extent in the future is more than 2.5 times that of the past.

### 4.3. E-Learning Workshops for Fostering "Facilitators"

Two e-learning workshops, introductory lectures in April 2021 and hands-on training in January 2022, were conducted to promote fostering Facilitators in Davao City. In line with the selection criteria produced in a co-designing manner, the workshops gathered about thirty candidates of Facilitators specializing in six different disciplines from national and local governments, academia, CSO and NGO, the private sector, and the media, as shown in Table 3. While the participants were instructed to download lecture materials from OSS-SR and learn them by themselves, the first workshop consisted of a month-long series of interactive sessions, such as an opening session, lecture introduction sessions, Q&A sessions, an examination, assignment submission, and a closing session. The hands-on training also included interactive sessions of an opening session, lecture introduction sessions, Q&A sessions, and a closing session. During the workshops, the participants were given assignments to produce deliverables for public dissemination, such as hazard maps considering climate change impact, local contingency plans, and action plans. The produced deliverables, which are twenty-one risk communication plans and workshop designs for fourteen barangays considering geographic, demographic, economic, and social features, can be particularly useful for the next actions, in which Facilitators will play the leading role in delivering the information to all levels of actors in society through workshops, seminars, training, focus group discussions, fact sheets, posters, and many other opportunities and tools.

**Table 3.** Disciplines of Facilitators who have participated in the e-learning workshops.

| Discipline | Number of Participants | |
| --- | --- | --- |
| | 1st Workshop for Introductory Lectures | 2nd Workshop for Hands-on Training |
| National government | 11 | 10 |
| Local government | 2 | 4 |
| Academe | 11 | 13 |
| Civil Society Organizations, Non-Governmental Organizations | 1 | 2 |
| Private sector | 2 | 1 |
| Media | 2 | 1 |
| TOTAL | 29 | 31 |

### 4.4. Dissemination by Facilitators

After the training through the e-learning workshops, Facilitators are expected to undertake the role of disseminating their learnings to improve disaster literacy in society as a whole and collect feedback from the public. However, since society consists of different actors, it is important to decide what content to deliver and what approach to take for

effective communication according to the target audiences. In Davao City, PLATFORM identified six target audience types by referring to the disciplines of Facilitators and co-designed the contents and communication methods considered effective for each target audience, as shown in Table 4. Although the contents and communication methods greatly depend on various aspects such as climatology, sociality, and geopolitics, this research was able to find a valid practice.

**Table 4.** Contents and methods of science communication with local society. Knowledge learned from OSS-SR and e-learning workshops is disseminated and translated in different ways according to the target audiences.

| Target Audience | Contents to Disseminate/Translate | Effective Communication Methods |
|---|---|---|
| Local community | - Causes of flood<br>- Climate change impacts<br>- Contingency planning for DRR | - Focus group discussions<br>- Radio/TV programs<br>- Posters |
| DRRM team | - Flood monitoring<br>- Flood hazard mapping<br>- Disaster risk management cycle | - Training<br>- Handouts |
| Government Agency | - Flood monitoring<br>- Flood hazard mapping<br>- Vertical and horizontal integration of DRRM plans and development plans | - Focus group discussions<br>- Fact sheets |
| Policymaker | - Causes of flood<br>- Climate change impacts<br>- Contingency planning for DRR<br>- Vertical and horizontal integration of DRRM plans and development plans | - Policy briefs<br>- Policy recommendations<br>- Fact sheets |
| Private sector | - Causes of flood<br>- Climate change impacts | - Fact sheets<br>- Posters |
| Media | - Causes of flood<br>- Climate change impacts | - Media releases |
| NGO and CSO | - Disaster risk management cycle<br>- Contingency planning for DRR | - Focus group discussions<br>- Handouts<br>- Posters |

## 5. Discussion

Although this paper successfully illustrates a practical implementation method to enhance flood resilience in a co-designing manner among relevant stakeholders, the applied method has not been quantitatively assessed for the improvement of flood resilience. Qualitative approaches account for 61% of studies on flood resilience [4]; nevertheless, the quantitative assessment of enhanced resilience will be demanded in case of comparison with other approaches. A study on quantitative evaluation of enhanced resilience by the hydrological model, agent-based model, or indicator should be further addressed.

The impact of climate change in Davao City was assessed using a comparison of the worst flood events in past and future climates, produced from a single GCM. For a more elaborate and reliable impact assessment with various flood magnitudes, a statistical method based on ensemble outputs from multiple GCMs is expected to be applied as a further assessment.

## 6. Conclusions

This study demonstrated a methodology for enhancing flood resilience by improving society-wide disaster literacy through the development of OSS-SR and the fostering Facilitators under the framework of PLATFORM in Davao City, Philippines. The OSS-SR development and the Facilitator training have been implemented through the co-design of governance in which all stakeholders engage. OSS-SR integrates knowledge, information, experiences, awareness, and models under different disciplines for consilience. Facilitators play a vital role, as a catalytic existence, in interlinking the scientific community and all levels of local society.

The localized OSS-SR for Davao City highlighted real-time flood forecasting and climate-change impact assessment, as well as the e-learning function. The development of two flood models with different spatial scales (basin and barangay scales) realized real-time flood forecasting, which can contribute to flooding risk mitigation by early warning, and climate change impact assessment, which can clarify the magnitude and frequency of future floods. In Davao City, increases in 24 h maximum rainfall, peak discharge, inundation extent, and inundation volume are confirmed at both basin and barangay scales in the case of the expected future climate. In particular, the simulation found that the basin-scale inundation area and the barangay-scale inundation volume under the expected future climate will be about four times as large as under the past climate.

Two e-learning workshops, which were co-designed among all levels of stakeholders, gathered about thirty candidates of Facilitators from six different disciplines according to the four selection criteria. During the e-learning workshops, the participants learned to skillfully operate OSS-SR and produced deliverables for public dissemination, such as hazard maps considering the impact of climate change, local contingency plans, and action plans. The deliverables, which comprise twenty-one risk communication plans and workshop designs for fourteen barangays considering geographic, demographic, economic, and social features, can be utilized at various opportunities at which Facilitators will take the initiative in deliver them to all levels of the target audiences in society. PLATFORM also co-designed the contents to disseminate and the communication methods that are effective for each target audience.

This study presented a practical method to enhance flood resilience, demonstrating the synthesis of science-based knowledge, such as the integration of different scales of hydrological models and climate-change impact assessment, and human resource development, such as fostering Facilitators capable of translating knowledge for communities, can fill the gaps between the science community and local society.

**Author Contributions:** Conceptualization, M.M., A.C.S. and T.K.; methodology, M.M., T.U., A.W.M.R., D.G.B., A.C.S. and T.K.; software, D.K., T.U., A.W.M.R., M.Y. and M.K.; formal analysis, M.M., D.K. and T.K.; investigation, M.M., D.G.B. and A.C.S.; resources, M.Y., A.C.S., T.K. and M.K.; data curation, M.M., D.K., A.W.M.R. and M.Y.; writing—original draft preparation, M.M.; writing—review and editing, D.K., T.U., A.W.M.R., D.G.B., A.C.S. and T.K.; visualization, M.M., D.K., M.Y. and M.K.; supervision, D.G.B., A.C.S. and T.K.; project administration, D.G.B., A.C.S. and T.K. All authors have read and agreed to the published version of the manuscript.

**Funding:** This research was financially supported by the Integrated Research Program for Advancing Climate Models (TOUGOU) Grant Number JPMXD0717935457 from the Ministry of Education, Culture, Sports, Science, and Technology (MEXT), Japan. Additionally, DIAS has been continuously received financial support from MEXT.

**Acknowledgments:** This research activity was supported by the Platform on Water Resilience and Disasters in the Philippines and the International Flood Initiative (IFI).

**Conflicts of Interest:** The authors declare no conflict of interest.

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
