# Peer review of "Co-Design for Enhancing Flood Resilience in Davao City, Philippines"

_water, doi:10.3390/w14060978_

Round 1

Reviewer 1 Report

Authors have made formidable efforts to enhancing flood resilience in Davao City, Philippines. Paper is rather well organized and written. However, the paper needs moderate modifications before it is processed: 

(1) Quantification of results can be merged into abstract section.

(2) Literature review is rather poorly-written. Use the most up-to-dated references:Flood monitoring by integration of Remote Sensing technique and Multi-Criteria Decision Making method, Water 13 (21), 3115.

(3) How was flood resilience computed in this study?? A new section is essential to add the revision.  

Reviewer 2 Report

The title of this paper is “Co-design for Enhancing Flood Resilience in Davao City, Philippines”. The title of this study is interesting. However, the contents of this study should be revised. Therefore, I recommend a major revision in this paper.

Round 2

Reviewer 1 Report

Accept as is

Reviewer 2 Report

The title of this paper is “Co-design for Enhancing Flood Resilience in Davao City, Philippines”. This paper was corrected according to the reviewer’s opinions. Therefore, I recommend an acceptance in this paper.
